# Maternal Serum Concentrations of Per- and Polyfluoroalkyl Substances in Early Pregnancy and Small for Gestational Age in Southern Sweden

**DOI:** 10.3390/toxics11090750

**Published:** 2023-09-04

**Authors:** Ellen Malm, Andreas Vilhelmsson, Hannah Högfeldt, Isabelle Deshayes, Karin Källén, Stefan R. Hansson, Christian H. Lindh, Lars Rylander

**Affiliations:** 1Division of Occupational and Environmental Medicine, Department of Laboratory Medicine, Lund University, MV (Building 402a), 223 81 Lund, Sweden; andreas.vilhelmsson@med.lu.se (A.V.); ha5566ho-s@student.lu.se (H.H.); isabelle.deshayes@outlook.com (I.D.); christian.lindh@med.lu.se (C.H.L.); lars.rylander@med.lu.se (L.R.); 2Tornblad Institute, Lund University, Biskopsgatan 7, 223 62 Lund, Sweden; karin.kallen@med.lu.se; 3Division of Obstetrics and Gynecology, Department of Clinical Sciences Lund, Lund University, 221 85 Lund, Sweden; stefan.hansson@med.lu.se

**Keywords:** SGA, PFAS, endocrine disrupting chemical, case control, biobank, fetal growth

## Abstract

Small for gestational age (SGA) is considered an adverse birth outcome. Per- and polyfluoralkyl substances (PFAS) have become increasingly investigated as contributing environmental factors, thus far with inconclusive results. The current study aimed to investigate the hypothesized association between increased maternal PFAS levels in early pregnancy and an increased risk for SGA birth. This population-based study used data from a sample of children born in Scania, Southern Sweden, between 1995 and 2009. Two groups were compared: cases born with SGA (*n* = 298) and non-SGA controls (*n* = 580). The cases consisted of two subgroups: one included women whose children’s growth in late pregnancy was in the lowest quartile, and another included women from the remaining growth quartiles. Corresponding maternal serum samples were collected from a biobank and analyzed for concentrations of four types of PFAS: perfluorooctanoic acid (PFOA), perfluorononanoic acid (PFNA), perfluorohexane sulfonic acid (PFHxS), and perfluorooctane sulfonic acid (PFOS) using liquid chromatography–tandem mass spectrometry (LC/MS/MS). The results were combined with information from birth registers and analyzed using Mann–Whitney U-tests and logistic regression—unadjusted as well as adjusted for potential confounders. In conclusion, elevated maternal concentrations of PFAS were not associated with an increased risk of SGA birth. However, significant ORs were observed in a subgroup analysis restricted to women of Nordic origin (unadjusted OR 3.2 and adjusted OR 2.4) for PFHxS.

## 1. Introduction

In recent years, per- and polyfluorinated alkyl substances (PFAS), a class of widely used, highly persistent, industrially produced compounds, have become a cause for environmental concern and are associated with adverse health outcomes [1].

PFAS are a vast and complex group of manmade substances classified as organic pollutants with half-lives of several years [2,3]. Their common properties include chemical and thermal stability, hydrophobia and lipophobia, the ability to reduce aqueous surface tension, and the ability to create stable foams. As a result of their unique chemical structures, PFAS are highly persistent and suitable for industrial use, for example, as surfactants. In addition to their extensive industrial applications, PFAS can be found in commercial products, e.g., non-stick cookware, oil-, stain-, and water-resistant coatings in clothing, carpets, personal protective equipment, fire-extinguishing foams, and cosmetic products.

PFAS are released and distributed to the environment through direct emissions during the production, use, and disposal of products in which PFAS are present [4]. The environmental impact is further increased by indirect emissions from the transformation of precursor substances—PFAS created through the degradation of compounds in other substances [5]. Human exposure occurs primarily via diet and drinking water due to surface and groundwater contamination [6]. Although the effects of PFAS have become increasingly investigated, the extent of their environmental and biological impact has yet to be determined. During gestation, a fetus is exposed to maternal PFAS via placental transportation and passage [7]. Strong correlations between maternal and cord serum PFAS concentrations have been observed, enabling maternal concentrations to be used as a surrogate marker for fetal exposure during pregnancy [8]. PFAS have been described as endocrine-disrupting chemicals (EDC), and exposure may affect fetal growth and development [9,10].

Birth weight is an established indicator of fetal development and healthy birth outcomes [11]. Small for gestational age (SGA) is defined as a baby born smaller than the average for a specific week of pregnancy. Morbidity and mortality are significantly increased in infants born SGA when compared with appropriate-for-gestational-age infants (AGA), even from term, low-risk pregnancies [12]. While causes for SGA are diverse, the most common etiologies include maternal medical conditions, infections, substance use, and a variety of placental factors [13]. Numerous studies have investigated the association between fetal intrauterine PFAS exposure and adverse birth outcomes, such as reduced birth weight and SGA, but findings are inconsistent.

The most recent opinion by the European Food Safety Authority (EFSA) concluded that “there may be a casual association between PFOS and PFOA and birth weight” [14]. In addition, a systematic review and meta-analysis observed a statistically significant association between PFAS and reduced birth weight for four of the most widespread and frequently investigated PFAS: perfluorooctanoic acid (PFOA), perfluorononanoic acid (PFNA), perfluorohexane sulfonic acid (PFHxS), and perfluorooctane sulfonic acid (PFOS), as well as other types [15]. However, the authors highlighted the need to investigate the effect of specific types and doses of PFAS on birth outcomes, expanding our understanding of causality and dose–response curves.

The present study aimed to investigate the hypothesized association between maternal levels of specific PFAS in early pregnancy and the risk of SGA birth by combining medical birth register data with serum samples from a biobank in Scania, the southernmost county in Sweden.

## 2. Materials and Methods

### 2.1. Study Design and Sources

This study was approved by the Ethical Committee (Reg. No. 2014/696) at Lund University, Sweden, and was conducted as a register- and biobank-based case-control study where the following sources were used:

#### 2.1.1. Biobank Serum Samples

This study was based in Scania, where the Southern Sweden Maternal Biobank (SSMB) has collected >250,000 maternal blood samples from infection screening in early pregnancy (weeks 10–14) since 1989. The coverage in the biobank has been shown to be >90% [16].

#### 2.1.2. Register Data

Various maternal and obstetric registers and databases were utilized: the Swedish Medical Birth Register (MBR) and the Perinatal Revision South (PRS), a local Scanian birth register. As part of the Scanian regional ultrasound routines, examinations in early (around gestational week 18) and late (around gestational week 32) pregnancy were performed and registered in a separate database [17,18].

### 2.2. Cases and Controls

An *a priori* statistical power analysis was performed to determine the study population size. Based on the aim of including 300 SGA cases and 600 controls, detecting an odds ratio of 1.6 or higher with statistical significance (*p* < 0.05) was deemed possible while not exceeding the project budget.

Figure 1 illustrates the study selection process. The study population consisted of all singleton births in Scania between 1995 and 2009, identified through the registers mentioned above. Mothers diagnosed with preeclampsia were excluded due to an ongoing study on the association between PFAS and preeclampsia [16].

The SGA diagnosis was defined as birth weight below two standard deviations from the expected, according to the national intrauterine growth curve, which takes gestational week and gender into account [19]. Fetal growth in late pregnancy, i.e., between the second ultrasound and birth, was calculated for all SGA children. Among the women whose children’s growth in late pregnancy was in the lowest quartile, 225 mothers were randomly selected. This case group was defined as “SGA with poor growth spurt”. Likewise, a group of 225 women was randomly selected from the remaining upper three quartiles, defined as “SGA other”. From the study population, 900 children with regular birth weights were randomly selected as controls.

Among the 225 SGA with a poor growth spurt, the 225 SGA other, and the 900 controls, 95%, 96%, and 95% serum samples were available in the biobank, respectively. The biobank was instructed to select samples with the aim of later reaching 150, 150, and 600 analyzed samples, respectively. After randomly sorting the individuals within the respective groups, the first 158 SGA with a poor growth spurt, 151 SGA other, and 603 controls were therefore selected for PFAS analysis—accounting for potential loss. The remaining samples were excluded from this study due to budgetary reasons. Although available in the biobank, not all samples met the criteria regarding sample volume to be analyzed for concentrations of PFAS. Six SGA with a poor growth spurt, five SGA other, and twenty-three controls could not be analyzed, and the final numbers of participants were 152 SGA with a poor growth spurt, 146 SGA other, and 580 controls.

### 2.3. Analysis of PFAS

The maternal serum samples were analyzed using liquid chromatography–tandem mass spectrometry (LC/MS/MS; QTRAP 5500; AB Sciex, Framingham, MA, USA) using a method by Norén et al. at the laboratory of Occupational and Environmental Medicine at Lund University [20]. In summary, internal standards for all compounds were added to aliquots of serum. The proteins were precipitated with acetonitrile, vigorously shaken for 30 min, and centrifuged and analyzed regarding concentrations of perfluorooctane sulfonic acid (PFOS), perfluorooctanoic acid (PFOA), perfluornonanoic acid (PFNA), and perfluorohexane sulfonic acid (PFHxS). All native and isotopically labeled standards were purchased from Wellington Laboratories (Guelph, ON, Canada). Acetonitrile, ammonium acetate, and methanol were from Merck (Darmstadt, Germany), and water was from a Milli-Q Integral 5 system (Millipore, Billerica, MA, USA). Homemade quality control (QC) samples were prepared by pooling serum samples, and MO water was used for chemically blank samples. Every analyzed batch included calibration standards, QC samples, and chemically blank samples. The limit of detection (LOD) was determined from the blank samples.

The analyses were conducted in a duplicate and randomized order with LODs of 0.12 ng/mL for PFOS, 0.04 ng/mL for PFOA, and 0.03 ng/mL for PFNA and PFHxS. The coefficient of variation (CV) of the QC samples (*n* = 32) was 8% for PFHxS at 2 ng/mL and 10% at 3 ng/mL, for PFOA, 12% at 3 ng/mL and 9% at 4 ng/mL; for PFNA 10% at 2 ng/mL and 9% at 4 ng/mL; and for PFOS 7% at 12 ng/mL and 8% at 13 ng/mL. The laboratory participated successfully in the HBM4EU QA/QC program and participates in G-EQUAS for PFOS and PFOA analysis coordinated by the University of Erlangen-Nuremberg, Germany (Appendix A).

### 2.4. Other Pregnancy Information

Additional information was obtained from the MBR, which includes data on practically all deliveries in Sweden with information acquired from medical records on prenatal, delivery, and neonatal care [21]. The following variables were obtained: maternal age, calendar year of birth, body mass index (BMI) kg/m^2^, parity (primipara and multipara), smoking habits (non-smoker, 1–9 cigarettes/day, and >9 cigarettes a day), sex of the child, diabetes (yes/no), gestational diabetes (yes/no), involuntary childlessness for at least a year (yes/no), gestational week at partus, and maternal country of origin (eight categories).

### 2.5. Statistics

Background characteristics and PFAS concentrations for the cases and controls are presented as mean/median with minimum and maximum values for continuous variables or numbers and percentages for categorical variables. As a first step, comparisons of PFAS concentrations as continuous variables between cases and controls were analyzed with the Mann–Whitney U-test. In addition to managing the cases as a single group, the cases were divided into two subgroups (SGA with poor growth spurt and SGA other) and analyzed separately. As a second step, odds ratios (ORs) and 95% confidence intervals (CIs) were calculated using logistic regression models. In these analyses, the concentrations of the PFAS variables were categorized into quartiles based on their distribution among the controls. The lowest exposure quartiles were used as reference categories. The PFAS variables were analyzed individually. Crude analyses were completed with *a priori* defined models adjusted for maternal age (4 categories: <25, 26–30, 31–35, and >35 years), BMI (4 categories: <20, 20-<25, 25-<30, ≥30 kg/m^2^), smoking (2 categories: yes and no), and parity (primipara and multipara). Based on prior research, women of Nordic origin have among the highest concentrations of PFAS; therefore, we conducted separate analyses including only women of Nordic origin [22]. In addition, it is reasonable to assume that these women are more homogeneous as a group compared to the entire study population. Furthermore, separate analyses were performed for infant sex, as previous studies have found sex-specific effects of PFAS on adverse birth outcomes [23,24]. Due to limited statistical power, separate analyses were not performed for the different case groups. Statistical significance was defined as *p*-values below 0.05. All statistical analyses were conducted in SPSS (Version 27.0) (IBM Corp., Armonk, NY, USA).

## 3. Results

Background characteristics and demographics of the study participants are presented in Table 1. The most pronounced differences between the cases and controls were the higher proportion of primiparas (66.8% vs. 47.1%) and smokers (23% vs. 9.6%), and the lower proportion of women originating from a Nordic country.

The highest maternal serum concentrations were observed for PFOS in all study groups, followed by PFOA (Table 2). There were no significant differences between all cases and the controls when the PFAS were analyzed as continuous variables. The cases with poor growth spurt had significantly higher concentrations of PFOA and PFHxS as compared to the controls, whereas the case group SGA other had lower concentrations of PFOS compared to the controls.

When the different PFAS were analyzed as categorical variables and when the highest vs. lowest exposure categories were compared, the only statistically significant associations observed were ORs below one in the adjusted analysis for PFOS when all cases were included (OR 0.60, 95% CI 0.38–0.96, Table 3). This observation was driven by the group SGA other (unadjusted OR 0.49, 95% CI 0.28–0.85; adjusted OR 0.40, 95% CI 0.21–0.75). For PFHxS, the unadjusted analyses gave a significantly increased OR of 2.00 (95% CI 1.21–3.30) for SGA poor growth spurt, which decreased to 1.51 (95% CI 0.83–2.74) in the adjusted analyses and was no longer statistically significant.

In the analyses including only mothers of Nordic origin, significantly increased ORs were observed when the highest exposure category was compared to the lowest exposure category for PFOA (OR 2.85 95% CI 1.43–5.67), PFOS (OR 1.77 95% CI 1.02–3.10), as well as for PFHxS (OR 3.19 95% CI 1.60–6.34) (Table 4). In the adjusted analyses, PFHxS was the only one still statistically significant (OR 2.44, 95% CI 1.11–5.38). When separate analyses were performed for infant sex, two significant associations were observed. A decreased OR among boys for PFOS (adjusted OR 0.38 95% CI 0.20–0.74) was observed when the highest exposure category was compared to the lowest exposure category, and a significantly increased OR among girls was observed for PFHxS (adjusted OR 2.08 95% CI 1.08–4.02).

## 4. Discussion

The present study showed no statistically significant increased risk of SGA birth with higher maternal serum concentrations of PFAS in the general population in Scania, Southern Sweden. Results indicating a reduced risk of SGA birth with higher levels of PFOA and PFOS were observed in some instances—however sporadic and inconsistent across exposure quartiles. In contrast, we found significant associations between exposure to PFHxS and an increased risk of SGA birth in the analysis restricted to women of Nordic origin.

### 4.1. Previous Studies 

Our main results are in agreement with recent EFSA opinions lacking strong support for the hypothesized association between increased PFAS exposure and SGA birth [14]. The most recent opinion reports a tendency among the examined studies towards an inverse correlation between both PFOS and PFOA and birth weight, while acknowledging the lack of strong indications that this translates into an increased risk of SGA birth.

The inconsistent and occasionally conflicting evidence from epidemiological studies on PFAS exposure and reduced birth weight is an ever-recurring predicament, perhaps further complicated by the physiological changes related to pregnancy. A recently updated meta- and bias-analysis on serum PFOA and birth weight found little or no association when exclusively including studies where blood was sampled from mothers in early pregnancy [15]. However, an association was observed when restricting the analysis to samples drawn later in pregnancy. The authors discuss a possible and repeatedly suggested explanation for the heterogeneity of the results: the timing of maternal blood sampling. More specifically, the differences may be attributed to a pregnancy-related expansion of plasma and blood volume and the parallel increase in glomerular filtration rate (GFR), which would be expected to result in reduced concentrations of PFOA later in pregnancy [25,26]. The authors argue that studies based on samples drawn in late pregnancy are more susceptible to confounding by low GFR [27]. An insufficient GFR during pregnancy leads to impaired fetal growth as well as higher maternal serum PFOA, which naturally would be more of a concern later in pregnancy as opposed to earlier.

### 4.2. Restricted Analyses 

The analyses restricted to male infants and women of Nordic origin were specifically selected based on results from previous studies. Numerous research articles on the subject have presented sex-specific effects of PFAS, with male fetuses being more affected regarding birth weight [23,24]. In our study, higher PFOS concentrations resulted in a decreased risk of SGA among boys, whereas higher PFHxS concentrations resulted in an increased risk of SGA among girls. The reason for this pattern remains unknown. Our study found significant differences when restricting the analysis to women of Nordic origin. It is reasonable to assume that Nordic women are more homogenous as a group compared to the entire study population, which might reduce the risk of residual confounding. In addition, epidemiological studies have found PFAS serum levels to be among the highest in these countries [22]. The highest maternal serum concentrations were found in women of Swedish and Danish origin, compared to study participants born elsewhere. Unfortunately, no record of resident duration was available, but the discrepancy was speculated to be a result of differences in background exposure, lifestyle, diet, and perhaps even genetic differences in susceptibility to exposure. Regarding the discrepancy between the ORs in the unadjusted and adjusted models, it was found to be driven by parity when adjusting for all confounders separately. This observation agrees with prior research, as parity has been found to be a significant predictor for PFAS concentrations, decreasing with parity increase [28].

### 4.3. Strengths and Weaknesses

The strengths of our study include the use of the Southern Sweden Maternal Biobank, extensive birth registers, and a population-based design, which increases generalizability. The high-quality registers enabled us to adjust for important potential confounders, such as smoking habits and parity. By selecting subjects from a regional screening program, the risk of selection bias was minimized, and ultrasound data allowed us to take infant growth in the last trimester into account. Also, all samples were taken at the same gestational age and analyzed using state-of-the-art laboratory equipment. As previously mentioned, PFAS are stable molecules with half-lives of several years, meaning the concentration in a single sample can characterize exposure in epidemiological studies with relative accuracy.

Several limitations should be acknowledged. We could not account for the mother’s educational level or socioeconomic status, which have been identified as key confounders. Furthermore, mothers born SGA are more likely to give birth to SGA children themselves, and maternal birth weight was not accounted for in our study. Pathophysiological mechanisms of intrauterine growth restriction (IUGR), namely infections, various maternal diseases, chromosomal aberration, immunization, and placenta-associated complications such as preeclampsia, were not accounted for. In addition, growth estimates based on ultrasound data are susceptible to operator-dependent differences as well as inherent random errors. This could potentially lead to a nondifferential misclassification and bias estimates towards the null. As pointed out in the introduction, thousands of specific molecules constitute the group of PFAS, and the current study chose to investigate only four of them. Although we selected the most widespread and frequently studied substances, this may limit the external validity and applicability of the results.

## 5. Conclusions

In conclusion, the present study found no association between higher maternal serum levels of PFAS in early pregnancy and an increased risk of SGA birth. However, an increased risk was observed for PFHxS when restricting the analysis to women of Nordic origin. This was a population- and biobank-based study using early-pregnancy sampling while also taking the fetal growth spurt in late pregnancy into account, which is, to our knowledge, a new approach. Future investigations of the potential fetotoxic effects of PFAS are needed to further our understanding of their biological impact on humans.

## Figures and Tables

**Figure 1 toxics-11-00750-f001:**
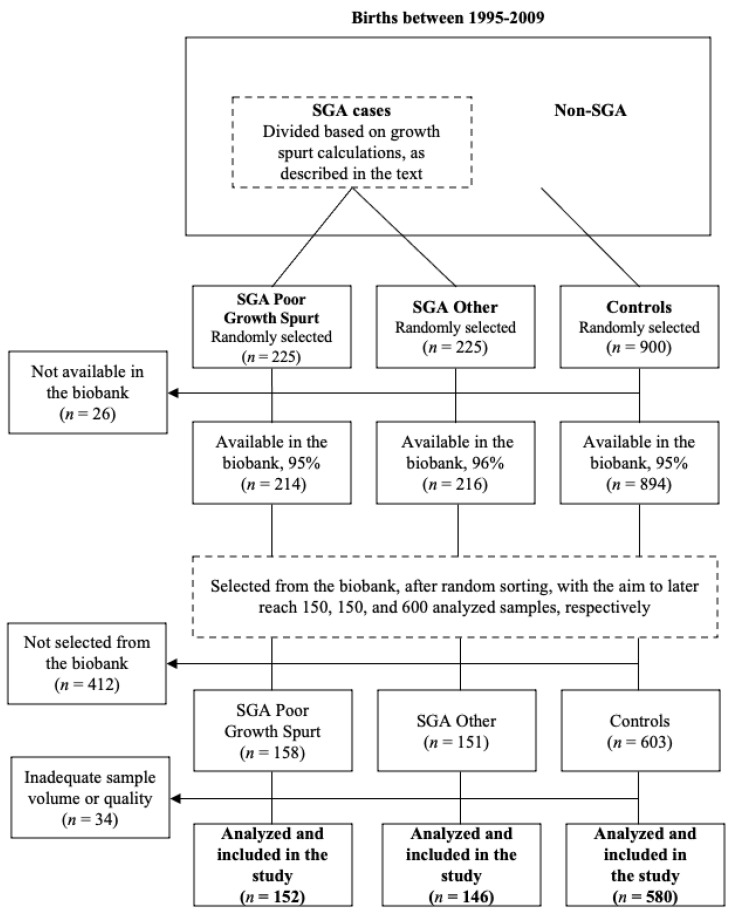
Flow diagram of the study selection process.

**Table 1 toxics-11-00750-t001:** Background characteristics among all cases (*n* = 298) and controls (*n* = 580) as well as for the two case groups.

Characteristics	Mean/Median (min, max)			
	SGA Poor Growth Spurt *n* = 152	SGA Other *n* = 146	Cases (Total) *n* = 298	Controls *n* = 580
**Maternal birth year**	1973/1973 (1956, 1988)	1974/1974 (1958, 1992)	1974/1973 (1956, 1992)	1973/1973 (1955, 1993)
**Maternal age at childbirth**	30/30 (19, 40)	28/29 (17, 41)	29/29 (17, 41)	29/29 (15, 42)
**Calendar-year of childbirth**	2003/2004 (1995, 2009)	2003/2004 (1995, 2009)	2003/2004 (1995, 2009)	2003/2004 (1995, 2009)
	*n* (%)			
**BMI in early pregnancy (kg/m^2^)**				
<20	11 (8.3)	24 (19.2)	35 (13.6)	60 (11.9)
20-<25	75 (56.8)	72 (57.6)	147 (57.2)	286 (56.9)
25-<30	25 (18.9)	21 (16.8)	46 (17.9)	109 (21.7)
≥30	21 (15.9)	8 (6.4)	29 11.3)	48 (9.5)
Missing	20	21	41	77
**Parity**				
1	103 (67.8)	96 (65.8)	199 (66.8)	273 (47.1)
≥2	49 (32.2)	50 (34.2)	99 (33.2)	307 (52.9)
**Smoking in early pregnancy**				
Non-smokers	106 (77.9)	105 (76.1)	211 (77.0)	490 (90.4)
1–9 cig/day	23 (16.9)	22 (15.9)	45 (16.4)	39 (7.2)
>9 cig/day	7 (5.1)	11 (8.0)	18 (6.6)	13 (2.4)
Missing	16	8	24	38
**Gender of child**				
Male	65 (42.8)	76 (52)	141 (47.3)	299 (51.6)
Female	87 (57.2)	70 (47.9)	157 (52.7)	281 (48.4)
**Diabetes (yes)**	0 (0.0)	1 (0.7)	1 (0.3)	2 (0.3)
**Gestational diabetes (yes)**	2 (1.3)	4 (2.7)	6 (2.0)	11 (1.9)
**Gestational week at partus**				
<34	1 (0.7)	0 (0.0)	1 (0.3)	2 (0.03)
34-<37	25 (16.4)	11 (7.5)	36 (12.1)	13 (2.2)
≥37	126 (82.9)	135 (92.5)	261 (87.6)	565 (97.4)
**Maternal country of origin**				
Sweden	97 (64.8)	70 (50.3)	167 (59.2)	369 (65.8)
Other Nordic Countries	6 (4.2)	<5	8 (2.8)	20 (3.6)
Western Europe, USA, Australia, New Zealand	<5	<5	<5	6 (1.2)
Former Eastern Europe	8 (5.6)	13 (9.3)	21 (7.4)	48 (8.6)
Sub-Saharan Africa	<5	<5	6 (2.1)	13 (2.3)
The Middle East, North Africa	23 (16.1)	37 (26.6)	60 (20.3)	76 (13.5)
East Asia	<5	12 (8.6)	16 (5.7)	23 (4.1)
South/Central America	<5	<5	<5	6 (1.2)
Missing	9	7	16	19

**Table 2 toxics-11-00750-t002:** Maternal serum concentrations of the different PFAS compounds (ng/mL) among the cases and controls. For comparison, only statistically significant (*p* < 0.05) differences between groups are marked.

PFAS (ng/mL)	SGA Poor Growth Spurt *n* = 152	SGA Other *n* = 146	Cases (Total) *n* = 298	Controls *n* = 580
	Mean/Median (min, max)			
**PFOA**	2.9 */2.9 (0.3, 10.1)	2.5/2.1 (0.4, 8.5)	2.7/2.6 (0.3, 10.1)	2.5/2.1 (0.4, 9.4)
**PFOS**	14.4/12.0 (1.1, 71.1)	12.6 */9.8 (0.6, 126.9)	13.5/10.8 (0.6, 126.9)	13.2/10.7 (0.5, 54.4
**PFNA**	0.5/0.4 (0.04, 2.8)	0.5/0.4 (0.04, 2.2)	0.5/0.4 (0.04, 2.8)	0.5/0.4 (0.03, 3.5)
**PFHxS**	0.8 */0.7 (<LOD, 9.4)	0.6/0.5 (<LOD, 9.7)	0.7/0.6 (<LOD, 9.7)	0.6/0.5 (<LOD, 4.5)

PFOA: perfluorooctanoic acid; PFOS: perfluorooctane sulfonate; PFNA: perfluorononanoic acid; and PFHxS: perfluorohexane sulfonate. Limit of detection (LOD): PFOA: 0.04, PFOS: 0.12, PFNA: 0.03, and PFHxS: 0.03. * Significant differences (*p* < 0.05) between cases and controls using Mann–Whitney U-test.

**Table 3 toxics-11-00750-t003:** The associations between concentrations of PFAS in maternal serum samples taken in early pregnancy and small for gestational age (SGA). Odds ratios (OR) and 95% confidence intervals (CI) were obtained from logistic regressions.

PFAS (ng/mL)	Cases and Controls (*n*)				Cases (Total)				SGA Poor Growth Spurt				SGA Other			
	SGA Poor Growth Spurt	SGA Other	SGA (Total)	Controls	Unadjusted		Adjusted ^a^		Unadjusted		Adjusted ^a^		Unadjusted		Adjusted ^a^	
					OR	95% CI	OR	95% CI	OR	95% CI	OR	95% CI	OR	95% CI	OR	95% CI
**PFOA**																
≤12.03	35	43	78	145	1.00	ref ^b^	1.00	ref ^b^	1.00	ref ^b^	1.00	ref ^b^	1.00	ref ^b^	1.00	ref ^b^
>12.03–19.62	16	27	43	145	**0.55**	**0.36–0.85**	**0.44**	**0.27–0.72**	**0.46**	**0.24–0.86**	**0.34**	**0.17–0.70**	0.63	0.37–1.07	**0.49**	**0.26–0.91**
>19.62–26.78	45	43	88	145	1.13	0.77–1.65	0.82	0.52–1.29	1.29	0.78–2.12	0.86	0.48–1.54	1.00	0.62–1.62	0.73	0.41–1.31
>26.78	56	33	89	145	1.14	0.78–1.67	0.75	0.47–1.19	1.60	0.99–2.59	0.93	0.52–1.65	0.77	0.46–1.28	0.55	0.30–1.01
**PFOS**																
≤6.66	40	47	87	145	1.00	ref ^b^	1.00	ref ^b^	1.00	ref ^b^	1.00	ref ^b^	1.00	ref ^b^	1.00	ref ^b^
>6.66–10.73	27	35	62	145	0.71	0.48–1.06	0.70	0.45–1.09	0.68	0.39–1.16	0.68	0.37–1.23	0.75	0.45–1.22	0.75	0.43–1.30
>10.73–18.09	47	41	88	145	1.01	0.70–1.47	0.87	0.57–1.35	1.18	0.73–1.90	0.99	0.57–1.73	0.87	0.54–1.41	0.78	0.45–1.35
>18.09	38	23	61	145	0.70	0.47–1.05	**0.60**	**0.38–0.96**	0.95	0.58–1.57	0.84	0.48–1.48	**0.49**	**0.28–0.85**	**0.40**	**0.21–0.75**
**PFNA**																
≤0.28	38	46	84	145	1.00	ref ^b^	1.00	ref ^b^	1.00	ref ^b^	1.00	ref ^b^	1.00	ref ^b^	1.00	ref ^b^
>0.28–0.41	27	34	61	145	0.73	0.49–1.09	0.65	0.41–1.03	0.71	0.41–1.23	0.61	0.33–1.11	0.74	0.45–1.22	0.67	0.37–1.19
>0.41–0.60	45	30	75	145	0.89	0.61–1.32	0.74	0.47–1.17	1.18	0.73–1.93	0.93	0.53–1.64	0.65	0.39–1.09	0.61	0.34–1.11
>0.60	42	36	78	145	0.93	0.63–1.36	0.82	0.52–1.29	1.11	0.67–1.81	0.88	0.49–1.58	0.78	0.48–1.28	0.78	0.44–1.40
**PFHxS**																
≤0.31	29	42	71	145	1.99	ref ^b^	1.00	ref ^b^	1.00	ref ^b^	1.00	ref ^b^	1.00	ref ^b^	1.00	ref ^b^
>0.31–0.53	32	37	69	145	0.97	0.65–1.46	0.97	0.62–1.52	1.10	0.64–1.92	1.13	0.61–2.07	0.88	0.54–1.45	0.81	0.46–1.43
>0.53–0.78	33	24	57	145	0.80	0.53–1.22	0.68	0.42–1.10	1.14	0.66–1.97	0.85	0.45–1.60	0.57	0.33–0.99	0.54	0.29–1.02
>0.78	58	43	101	145	1.42	0.97–2.08	1.16	0.74–1.84	**2.00**	**1.21–3.30**	1.51	0.83–2.74	1.02	0.63–1.66	0.92	0.51–1.64

PFOA: perfluorooctanoic acid; PFOS: perfluorooctane sulfonate; PFNA: perfluorononanoic acid; and PFHxS: perfluorohexane sulfonate. The PFAS variables were included in the models one at a time. ^a^ Adjusted for BMI (4 categories: <20, 20-<25, 25-<30, >30), parity (2 categories: 1 and ≥2), maternal smoking habits in early pregnancy (2 categories: yes or no), and age (4 categories: <20, 20-<25, 25-<30, 30). ^b^ Reference category.

**Table 4 toxics-11-00750-t004:** The associations between concentrations of PFAS in maternal serum samples taken in early pregnancy and small for gestational age (SGA), including only women of Nordic origin in the analysis. Odds ratios (OR) and 95% confidence intervals (CI) were obtained from logistic regressions.

PFAS (ng/mL)	Women of Nordic Origin			
	Unadjusted		Adjusted ^a^	
	OR	95% CI	OR	95% CI
**PFOA**				
≤12.03	1.00	ref ^b^	1.00	ref ^b^
>12.03–19.62	0.68	0.30–1.52	0.48	0.19–1.17
>19.62–26.78	**2.89**	**1.44–5.80**	1.87	0.85–4.12
>26.78	**2.85**	**1.43–5.67**	1.62	0.74–3.57
**PFOS**				
≤6.66	1.00	ref ^b^	1.00	ref ^b^
>6.66–10.73	1.00	0.54–1.84	0.92	0.46–1.85
>10.73–18.09	**1.77**	**1.02–3.10**	1.55	0.83–2.90
>18.09	1.33	0.76–2.31	1.11	0.59–2.09
**PFNA**				
≤0.28	1.00	ref ^b^	1.00	ref ^b^
>0.28–0.41	0.99	0.56–1.75	0.87	0.44–1.71
>0.41–0.60	1.17	0.67–2.05	1.14	0.59–2.20
>0.60	1.39	0.80–2.41	1.34	0.69–2.60
**PFHxS**				
≤0.31	1.00	ref^b^	1.00	ref ^b^
>0.31–0.53	1.69	0.82–3.48	1.68	0.75–3.77
>0.53–0.78	1.67	0.82–3.43	1.46	0.65–3.28
>0.78	**3.19**	**1.60–6.34**	**2.44**	**1.11–5.38**

PFOA: perfluorooctanoic acid; PFOS: perfluorooctane sulfonate; PFNA: perfluorononanoic acid; and PFHxS: perfluorohexane sulfonate. The PFAS variables were included in the models one at a time. ^a^ Adjusted for BMI (4 categories: <20, 20-<25, 25-<30, >30), parity (2 categories: 1 and ≥2), maternal smoking habits in early pregnancy (2 categories: yes or no), and age (4 categories: <20, 20-<25, 25-<30, 30). ^b^ Reference category.

## Data Availability

Due to the personal data collected and potentially identifying information contained within the data, the data are available upon request. Ethical approval by the Swedish Ethical Review Authority may be necessary, and requests for data may be sent to LUPOP—Lund University Population Research Platform via email (lupop@ed.lu.se).

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
