# Peer review of "Maternal Serum Concentrations of Per- and Polyfluoroalkyl Substances in Early Pregnancy and Small for Gestational Age in Southern Sweden"

_toxics, 2023, doi:10.3390/toxics11090750_

Round 1

Reviewer 1 Report

I have carefully read the manuscript describing connection between maternal serum concentrations of Per- and Polyfluoroalkyl substances in early pregnancy and small for gestational age (SGA) in southern Sweden. Authors presented an interested study with original results about association of PFAS concentration in early pregnancy of Sweden woman and risk of SGA birth. Since EFSA has already in 2018 recognized association between these substances and low birth weight, but with uncertainty, this investigation is valuable because it can help expanding knowledge and bringing new conclusions about effects of PFAS chemicals on the birth weight and the children’s health. The manuscript is well organized, results are clearly presented and references are up-to-date. Although the experimental design is clear, there are some concerns that need editing and corrections especially the Conclusion paragraph. I consider study reported in this manuscript applicable for publication after major revision.

Specific suggestions and comments are given below:

1)      In Materials and Methods section:

-          2.3. Manufacturers’ names and location for chemicals are missing.

-          2.3. What were the chemically blank samples?

-          It does not say nowhere that analyses were done on male offspring and in the Results section you mention that.

2)      In Results section:

-          Table 2 – in table description abbreviation for detection limit is wrong

-          Table 3

-          in the description of the results above the table for the group total cases it is not written is it for adjusted or unadjusted data

-          in the description of the results above the table for the group SGA other values for 95% CI is missing and there are no values for unadjusted analysis

-          in the last sentence where the results for PFHxS are described there is no information to which group it refers to

-          in the Table the minus sign in front of the numbers in the first row of every chemical is confused, maybe it should be written ≥, I am not sure what it means

-          Table 4

-          in the description of the results above the table the data for OR and 95%CI from the table should be written equally as in the table

-          if the analyses were made on male offspring the results should be shown

3)      In Discussion section:

It is necessary to emphasize in the Conclusion why is this study important, which is the novelty and what you did for the first time in this investigation. I suppose that is the timing of the sample collection that was in the early pregnancy. You should also explain why is this important.

4)      References number 14 and 25 are the same.

In the line 237 Reference number 16 is unappropriate.

The English Language should be revised by  a native speaker. 

Reviewer 2 Report

This is a well written and well designed study. However, I have some comments and minor suggestions for increasing the value of the paper.

The two different groups, cases and controls, differ significantly as declared by the authors in parity and smoking habits. Both these two factors are known to influence the levels of PFAS (and derivatives). Fabelova et al 2023, for instance decribe that primipara birth is associated to higher levels of PFAS. Santos et al 2021 describe that women exposed to passive smoking have lower levels of PFOA. I would like to see, in the discussion, how the differences between the groups can make differences in levels of PFAS-dervaitves and, if so, how that could affect the conclusion? 

I don´t understand table 3. In the control column there are 145 subjects for all different levels of PFAS-derivative?

At last, I can understand that, from a budget perspective, it is not always possible to perform all analysis that you want to do. However, in the discussion, section 4.2, it is clearly stated that, based on previous studies, the restricted analysis of male infants and women of Nordic origin was extra interesting. In figure 1 it is shown how the study population was extracted. Instead of random sampling from the biobank, why did you not collect only male infants and women of Nordic origin to increase power?

Round 2

Reviewer 1 Report

I have carefully read the reviewed version of the manuscript entitled “Maternal Serum Concentrations of Per- and Polyfluoroalkyl Substances in Early Pregnancy and Small for Gestational Age in Southern Sweden”.  Although you answered all questions and corrected the manuscript there are some things that are still not clear and few things that should be corrected. Also, the editing of the English language is necessary.

·         In Materials and Methods section

2.3. In the sentences you added in the text:

 ‘Homemade quality control (QC) samples were prepared by pooling serum samples and chemically blank samples were prepared by omitting serum and exchanged with MQ water. Every analyzed batch included calibration standards, QC samples, as well as chemically blank samples. The limit of detection (LOD) was defined from the blank samples.’

It is not necessary to write that ”… chemically blank samples were prepared by omitting serum and exchanged with MQ water, it is enough to write that chemically blank samples were prepared with MQ water.

Also I would like to ask how you did the calibration curve, with blood serum or with MQ water?

·         In Table 2

·         The abbreviations of PFAS compounds under the table should be written in the same order as they appeared in the table

·         There is no explanation for LOD abbreviation under the table

·         LOD values for PFAS compounds should be also written in the same order as they appeared in the table

·         It is not necessary that the sentence “All values were above LODs, except for PFHxS in some instances.” is written under the table.

·         Marks a and b should not be written by the abbreviations of PFAS compounds in the table. It is enough that the explanations are under the table.

·         Instead of “Exposure” in the first line it should be written PFAS as in the Table 3 and 4

·         The order of the PFAS compounds in the first column should be the same as in the Table 3 and 4.

In Table 3

·         I am not sure that you corrected the minus sign accurately, I think it should be instead of sign.

·         In the title of the Table you wrote “…and child’s birth weight (SGA)” that is not correctly because SGA in not the same as child’s birth weight but it is adverse birth outcome.

·         Marks a and b should not be written by the abbreviations of PFAS compounds in the table. It is enough that the explanations are under the table.

In Table 4

·         The minus sign should be correct as in the Table 3.

·         In the title of the Table the same thing should be corrected as in the Table 3.

·         Marks a and b should not be written by the abbreviations of PFAS compounds in the table. It is enough that the explanations are under the table.

·         The reference number 20 (line 131) is wrongly written in the text. It is not the same as the reference in the literature list (Lindh et al 2012 instead of Noren et al).

·         In the Statistics section you mentioned that “…separate analyses were performed for infant sex”, than in the Results section you wrote that “In the analyses including only male offspring, no statistically significant associations were observed.” but you did not mention the female offspring and in the Conclusion you wrote that “…Numerous research articles on the subject have presented sex-specific effects of PFAS, with male fetuses being more affected regarding concerning birth weight [23,24]. This was not the case in our study.”. My question is how you make this conclusion if you did not investigate the both offspring sexes?

The editing of the English language is necessary.
